# The Functional Characteristics and Soluble Expression of Saffron *Cs*CCD2

**DOI:** 10.3390/ijms242015090

**Published:** 2023-10-11

**Authors:** Ying Wang, Siqi Li, Ze Zhou, Lifen Sun, Jing Sun, Chuanpu Shen, Ranran Gao, Jingyuan Song, Xiangdong Pu

**Affiliations:** 1Inflammation and Immune Mediated Diseases Laboratory of Anhui Province, Anhui Institute of Innovative Drugs, School of Pharmacy, Anhui Medical University, Hefei 230032, China; 2Center of Traditional Chinese Medicine Formula Granule, Anhui Medical University, Hefei 230032, China; 3Institute of Chinese Materia Medica, China Academy of Chinese Medical Sciences, Beijing 100700, China; 4Key Lab of Chinese Medicine Resources Conservation, State Administration of Traditional Chinese Medicine of the People’s Republic of China, Institute of Medicinal Plant Development, Chinese Academy of Medical Sciences & Peking Union Medical College, Beijing 100193, China

**Keywords:** crocins, *Cs*CCD2, saffron, soluble expression, fusion tag

## Abstract

Crocins are important natural products predominantly obtained from the stigma of saffron, and that can be utilized as a medicinal compound, spice, and colorant with significant promise in the pharmaceutical, food, and cosmetic industries. Carotenoid cleavage dioxygenase 2 (*Cs*CCD2) is a crucial limiting enzyme that has been reported to be responsible for the cleavage of zeaxanthin in the crocin biosynthetic pathway. However, the catalytic activity of *Cs*CCD2 on β-carotene/lycopene remains elusive, and the soluble expression of *Cs*CCD2 remains a big challenge. In this study, we reported the functional characteristics of *Cs*CCD2, that can catalyze not only zeaxanthin cleavage but also β-carotene and lycopene cleavage. The molecular basis of the divergent functionality of *Cs*CCD2 was elucidated using bioinformatic analysis and truncation studies. The protein expression optimization results demonstrated that the use of a maltose-binding protein (MBP) tag and the optimization of the induction conditions resulted in the production of more soluble protein. Correspondingly, the catalytic efficiency of soluble *Cs*CCD2 was higher than that of the insoluble one, and the results further validated its functional verification. This study not only broadened the substrate profile of *Cs*CCD2, but also achieved the soluble expression of *Cs*CCD2. It provides a firm platform for *Cs*CCD2 crystal structure resolution and facilitates the synthesis of crocetin and crocins.

## 1. Introduction

Crocins, which are highly beneficial as medicines for human disorders and spices for flavoring and coloring, only accumulate in specific tissues, such as the stigmas of saffron (*Crocus sativus*), the fruits of *Gardenia jasminoides*, and the flowers of *Buddleja davidii*, of a few distantly related plants [1]. Numerous pharmacological studies have demonstrated that crocins exhibit several anticancer [2], anti-oxidative [3], anti-apoptotic [4], and anti-inflammatory effects [5]. Additionally, recent studies have reported the beneficial effects of crocins on Alzheimer’s disease and depression [6,7]. Saffron is the primary natural source of crocins, which confer to its stigma a characteristic dark red color [8]. Owing to their complicated isolation from plants and chemical synthesis, crocins command high market prices [9]. Crocins’ high price and remarkable properties in the treatment of central nervous system and cardiovascular diseases have led numerous scientists to investigate their biosynthetic pathway and heterologous production. With the rapid advancement in synthetic biology, the research on the synthesis of plant natural products using microbial fermentation has significantly progressed, overcoming the limitations of plant resources and providing a new route for the green and efficient production of plant natural compounds [10], such as artemisinic acid [11,12], etoposide aglycone [13], and cannabinoids [14]. Consequently, the heterologous synthesis of crocins using microorganisms is a significant addition to the existing production methods. The biosynthetic pathway of crocins includes the cleavage of carotenoids by carotenoid cleavage dioxygenases (CCDs), oxidation of aldehydes by aldehyde dehydrogenases (ALDHs), and transfer of glycosyl groups by UDP-glucuronosyltransferases (UGTs) (Figure 1) [1,8,15,16,17]. Among them, CCDs are considered to be rate-limiting enzymes in the biosynthetic pathway of crocins. CCDs, which are essential enzymes that catalyze carotenoid cleavage, are classified into four subfamilies—CCD1, CCD4, CCD7, and CCD8 [18]. The CCD1 subfamily is responsible for the production of volatile terpenoids, which are crucial for the formation of plant aroma [19]; the CCD4 subfamily is crucial for plant color formation [20,21,22]. CCD7 and CCD8 participate in the synthesis of the plant hormone strigolactone and play an important role in the germination of lateral roots and lateral buds [23]. Thus, the screening and identification of CCDs responsible for the synthesis of crocin precursors are crucial areas of research.

In 2014, Frusciantea et al. reported a novel key enzyme *Cs*CCD2 from saffron, which is considered to be the first key enzyme in the biosynthetic pathway of crocins [8]. It catalyzed the cleavage of zeaxanthin to produce crocetin dialdehyde but did not use β-carotene and lycopene as substrates. To further increase catalytic efficiency, researchers have engineered *Cs*CCD2 variants with broader substrate profiles for crocin biosynthesis using a “hybrid-tunnel” strategy. Based on the results of a directed evolution study, *Cs*CCD2^S323A^ was reported to have additional catalytic activity on β-carotene and improved catalytic efficiency [24]. Despite having better catalytic efficiency than the wild-type, the yield of the engineered strain remains in the milligram range and needs further optimization. In contrast, heterologous protein expression of CCDs is a significant challenge that impedes the discovery of molecular and physiological functions to some extent [25]. The low catalytic efficiency of *Cs*CCD2 may be attributed to its misfolding during heterologous expression in *E. coli*. However, a protein expression optimization study of *Cs*CCD2 remains missing.

The biosynthetic pathway of carotenoids in *Erwinia* species has been precisely identified [26]. crtE (GGPP synthase) can catalyze farnesyl pyrophosphate to form geranylgeranyl pyrophosphate (GGPP). crtB (PPPP synthase) is responsible for phytoene formation from GGPP, and phytoene is catalyzed by crtI (phytoene desaturase) to form lycopene. crtY (lycopene cyclase) is involved in β-carotene production from lycopene, and crtZ (β-carotene hydroxylase) can transfer β-carotene to zeaxanthin. Three plasmids were constructed based on these genes and used in carotenoid biosynthesis in *E. coli*. *E. coli* with pACCAR25ΔcrtX, which contained *crtE*, *crtI*, *crtB*, *crtY*, and *crtZ* from *E. uredovora*, was used in zeaxanthin production. *E. coli* with pACCAR16Δcrt harboring *crtE*, *crtI*, *crtB*, and *crtY* from *E. uredovora* can produce β-carotene. *E. coli* with pACCRT-EIB, which comprises *crtE*, *crtI,* and *crtB* from *E. uredovora*, is responsible for lycopene accumulation [26]. It establishes a solid foundation for the in vivo functional identification of the *CCD* gene.

In this study, we constructed the *Cs*CCD2 gene in a pET32a prokaryotic expression vector, and three engineered *E. coli* harboring pACCAR25ΔcrtX, pACCAR16Δcrt, and pACCRT-EIB were used for functional identification via in vivo assays [26]. Using ultra-performance liquid chromatography (UPLC) and UPLC with tandem mass spectrometry (UPLC-MS/MS), we demonstrated that the wild *Cs*CCD2 not only exhibited a catalytic impact on zeaxanthin but also possessed cleavage activity for β-carotene and lycopene. The molecular basis of the varied functionality of *Cs*CCD2 was also elucidated via a truncation study. Furthermore, a protein expression optimization study of *Cs*CCD2 revealed that maltose-binding protein (MBP)-*Cs*CCD2 was soluble using *E. coli* as the host organism at an induction temperature of 16 °C with 0.8 mM isopropyl-beta-D-thiogalactopyranoside (IPTG). Additionally, the catalytic efficiency of *Cs*CCD2 was enhanced as protein solubility increased. This will establish the groundwork for further research into the catalytic mechanism of *Cs*CCD2 and provide important genetic tools for the synthesis of crocetin and crocins.

## 2. Results

### 2.1. Functional Characteristics of CsCCD2

The prokaryotic expression vector TRX-*Cs*CCD2 (Table 1, Figure 2A) was constructed and co-transformed with pACCAR25ΔcrtX, pACCRT-EIB, or pACCAR16Δcrt into *E. coli* BL21(DE3) to create the TRX-CsCCD2-Z/L/B engineered strains. The pET32a vector (Table 1, Figure 2A) was also co-transformed with these three engineered plasmids to form pET32a-Z/L/B as a control. Following IPTG induction, the products were extracted and identified using UPLC and UPLC-MS/MS. The characteristic spectrums of crocetin dialdehyde (with mainly two maximum absorptions at 443.90 nm and 467.01 nm), zeaxanthin (with mainly two maximum absorptions at 450.30 nm and 473.90 nm), lycopene (with mainly three maximum absorptions at 446.71 nm, 468.47 nm, and 498.65 nm), and β-carotene (with mainly two maximum absorptions at 449.57 nm and 474.39 nm) are demonstrated in Appendix A, respectively. The UPLC results depicted that the extracts of TRX-*Cs*CCD2-Z/L/B produced a new peak at 14.55 min in accordance with the retention time of the crocetin dialdehyde standard. The characteristic spectrum of the new peak was parallel to the crocetin dialdehyde standard. Based on the UPLC-MS/MS analysis, the m/z of the new peak was 297.1855, which also corresponded to the crocetin dialdehyde standard (Figure 3C). Moreover, the fragmentation pattern of this new peak was consistent with the standard. However, no crocetin dialdehyde product was detected in pET32a-Z/L/B. These results revealed that TRX-*Cs*CCD2 cleaved zeaxanthin, β-carotene, and lycopene.

The function of DAN1-*Cs*CCD2, was identified again in this study using the same methodology as previously reported [8]. The prokaryotic expression vector DAN1-*Cs*CCD2 was co-transformed with pACCAR25ΔcrtX, pACCRT-EIB, or pACCAR16Δcrt into *E. coli* BL21(DE3) to construct the DAN1-CsCCD2-Z/L/B engineered strains. The pTHIO-DAN1 vector was co-transformed with these three engineered plasmids to form pTHIO-DAN1-Z/L/B as a control (Table 1). After being induced by arabinose, the products were extracted and tested using UPLC and UPLC-MS/MS. The UPLC results indicated that only DAN1-CsCCD2-Z exhibited a chromatographic peak with the same retention time as the crocetin dialdehyde standard. Mass spectrometry further confirmed that the product was the substance crocetin dialdehyde. These results demonstrated that DAN1-CsCCD2 only exhibited catalytic activity against zeaxanthin and had no activity on β-carotene or lycopene, which was consistent with the results reported in the literature (Figure 3B).

### 2.2. Optimization of CsCCD2 Protein Expression

The low soluble expression of *Cs*CCD2 may be responsible for its poor activity or inactivation. To enhance the soluble expression of *Cs*CCD2, three different fusion expression plasmids were constructed, including GST-*Cs*CCD2, SUMO-*Cs*CCD2, and MBP-*Cs*CCD2 (Figure 2B–D). Additionally, different induction temperatures and IPTG concentrations were tested. Following 24 h of induction with 0.5 mM IPTG, the *Cs*CCD2 proteins were successfully expressed among the three vectors and displayed as 96 kDa, 75 kDa, and 104 kDa protein bands on sodium dodecyl-sulfate polyacrylamide gel electrophoresis (SDS-PAGE), respectively. At a 37 °C induction temperature, *Cs*CCD2 protein was only detected in the inclusion body section of all three protein expression patterns (Appendix A). At a 28 °C induction temperature, *Cs*CCD2 protein was only detected in the inclusion body section of GST-*Cs*CCD2 and SUMO-*Cs*CCD2 expression patterns; however, it exhibited slightly soluble expression in MBP-*Cs*CCD2 (Appendix A). When the induced temperature decreased to 16 °C, *Cs*CCD2 was only detected in the inclusion body section of SUMO-*Cs*CCD2. Notably, in GST-*Cs*CCD2 and MBP-*Cs*CCD2, *Cs*CCD2 was expressed with more soluble protein. The SDS-PAGE results demonstrated that MBP-*Cs*CCD2 had better solubility than GST-*Cs*CCD2 (Figure 2E,F). It can be inferred that a lower induction temperature is more suitable for CsCCD2 expression. Furthermore, the results of induction with different IPTG concentrations on MBP-*Cs*CCD2 at 16 °C indicated that 0.8 mM was the best concentration for *Cs*CCD2 expression (Appendix A). In conclusion, the best soluble expression pattern of *Cs*CCD2 was obtained from MBP-*Cs*CCD2 at an induction temperature of 16 °C with a 0.8 mM IPTG concentration for 24 h.

### 2.3. The Catalytic Activity Study of CsCCD2-Fusion Protein

To examine the catalytic activity of *Cs*CCD2-fusion protein, GST-*Cs*CCD2, SUMO-*Cs*CCD2, and MBP-*Cs*CCD2 were constructed and transformed into *E. coli* BL21 (DE3) harboring pACCAR25ΔcrtX, pACCAR16Δcrt, or pACCRT-EIB to form GST/SUMO/MBP-*Cs*CCD2-Z/B/L engineered strains, respectively. After induction at 16 °C for 24 h, the UPLC results revealed that all engineered strains produced crocetin dialdehyde but with varying crocetin dialdehyde yields. MBP-*Cs*CCD2 exhibited higher conversion efficiency than GST-*Cs*CCD2 and SUMO-*Cs*CCD2 in vivo, which was consistent with the results of protein expression (Figure 4A–C). Thus, the use of the MBP tag can greatly improve the soluble expression of *Cs*CCD2 and result in higher CsCCD2 enzyme activity in the engineered *E. coli*.

### 2.4. Transforming the Catalytic Activity of CsCCD2 Using Tailored Truncation

In order to better understand the catalytic difference between TRX-*Cs*CCD2 and DAN1-*Cs*CCD2, a comparative analysis of the open reading frames (ORFs) of *Cs*CCD2, TRX-*Cs*CCD2, and DAN1-*Cs*CCD2 was conducted. There were two distinct variances between them, namely sections I and II (Figure 5). To the best of our knowledge, the redundant or missing amino acid sequence can have a significant impact on the three-dimensional structure of proteins, which in turn affects protein function. Thus, we truncated section II of DAN1-*Cs*CCD2 and obtained DAN1-*Cs*CCD2-T mutant (Figure 6A). Following functional characteristic analysis, DAN1-*Cs*CCD2-T indicated cleavable activity toward β-carotene and lycopene in addition to zeaxanthin. Furthermore, DAN1-*Cs*CCD2-T exhibited greater conversion efficiency than that observed before truncation (Figure 6B,C). This result indicated that the redundant amino acid sequences at the carbon-terminal end of CsCCD2 had a remarkable influence on its catalytic activity.

## 3. Discussion

Crocins, a class of highly valuable apocarotenoids derived primarily from saffron, have significant pharmacological activity for treating human disorders [28]. *Cs*CCD2 is the rate-limiting enzyme involved in the biosynthetic pathway of crocins in saffron. It has been reported that *Cs*CCD2 can only cleave the 7,8 and 7’,8’ double bonds of zeaxanthin sequentially to produce crocetin dialdehyde, but not without the cleavage of β-carotene and lycopene [8]. In this study, we replicated the functional assay of *Cs*CCD2 and obtained results consistent with those of previous investigations. Further research revealed that MBP-*Cs*CCD2 can effectively express its soluble form using *E. coli* as the host organism. Intriguingly, beta-carotene and lycopene, previously believed to be non-recognizable as substrates, were indeed accepted as substrates by *Cs*CCD2 in this study. What caused this difference?

First, in this study, the *Cs*CCD2 was subcloned into a pET32a prokaryotic expression vector to form TRX-*Cs*CCD2. The ORF in TRX-*Cs*CCD2 was TRXA•Tag-6×His-thrombin-S•Tag, which exhibited cleavage activity of zeaxanthin, β-carotene, and lycopene. However, in a previous report, researchers used the pTHIO-DAN1 prokaryotic expression vector to examine the functional characteristics of *Cs*CCD2. pTHIO-DAN1 is a derivative of pBAD/Thio (Invitrogen, Paisley, UK) carrying the pUC18 polylinker [27]. The ORF in DAN1-*Cs*CCD2 was HP-thioredoxin-enterokinase-*Cs*CCD2-V5•tag-6×His. The distinct open expression frame may account for the divergent functionality of *Cs*CCD2.

Second, to explain the divergent functionality of *Cs*CCD2 in the aforementioned two plasmids, an amino acid sequence comparative analysis was conducted, demonstrating two different sections between them, namely section I and section II (Figure 5). Thus, we hypothesized whether the amino acid sequence redundancy and deletion caused functional divergence. To verify this hypothesis, a DAN1-*Cs*CCD2 without section II was constructed, namely DAN1-*Cs*CCD2-T. Functional identification demonstrated that DAN1-*Cs*CCD2-T exhibited splitting activity on zeaxanthin, β-carotene, and lycopene. To some extent, the truncation study shed light on the molecular mechanism of functional divergence. This further suggests that using wild-type genes and thus avoiding additional modification sequences is better for conducting functional gene identification studies.

Third, as an important enzyme involved in the carotenoid cleavage process, *Cs*CCD2 exhibits relatively high specificity in substrate recognition and catalytic activity. The low catalytic efficiency of *Cs*CCD2 may also be attributed to its insoluble expression in *E. coli*. Thus, it is critical to investigate the soluble protein expression of *Cs*CCD2. Furthermore, a crystallographic structure analysis is the most direct and efficient way to gain a better understanding of the molecular catalytic mechanism of *Cs*CCD2. To the best of our knowledge, the crystal structures of only three CCD family members have been determined, including SynACO (4OU9) in *Synechocystis* sp. PCC 6803 [29], viviparous-14 (3NPE) in maize (*Zea mays*) [30], and NdCCD (6VCF) in *Nitrosotalea devanaterra* [25]. However, the crystal structure analysis of the CCD subfamily in the plant kingdom remains a significant challenge, and the insoluble expression of plant CCD limits its research progress. 

Fusion expression is one of the most effective strategies, which can either increase the recombinant protein expression or participate in the protein-folding process, to enhance the soluble expression of recombinant protein [31,32]. Many studies have demonstrated that some highly soluble proteins promoted the soluble expression of fusion proteins after fusion [33]. GST and MBP are the most commonly used fusion tags that can improve fusion-protein solubility and enable one-step purification via affinity chromatography [34,35]. Furthermore, SUMO can improve protein solubility by facilitating proper protein folding and enhancing binding stability [36]. *Cs*CCD2 fusion proteins with three different fusion tags (GST, MBP, and SUMO) were constructed using the fusion expression technique, including GST-*Cs*CCD2, SUMO-*Cs*CCD2, and MBP-*Cs*CCD2. The optimization of protein expression demonstrated that MBP-*Cs*CCD2 achieved soluble expression at an induction temperature of 16 °C with 0.8 mM IPTG. Concurrently, the catalytic efficiency was enhanced as the protein solubility increased. This breakthrough in the soluble expression of *Cs*CCD2 will greatly promote the progress of CCD crystallographic structure studies in the plant kingdom.

## 4. Materials and Methods

### 4.1. Chemicals and Strains

Standards of lycopene (CAS, 502-65-8), β-carotene (CAS, 7235-40-7), zeaxanthin (144-68-3), and crocetin dialdehyde (CAS, 502-70-5) were purchased from Sigma-Aldrich (Sigma-Aldrich Corp., St. Louis, MO, USA). Phanta Max Super-Fidelity DNA Polymerase was purchased from Vazyme (Nanjing Vazyme Biotech Co., Ltd., Nanjing, China). A ClonExpress II One Step Cloning Kit C112 was purchased from Vazyme. IPTG was purchased from Solarbio (Beijing Solarbio Science & Technology Co., Ltd., Beijing, China). All restriction endonucleases were purchased from Takara (Takara Biomedical Technology (Beijing) Co., Ltd., Beijing, China). T4 DNA Ligase was purchased from Takara. A DNA plasmid isolation kit and DNA gel extraction kit were purchased from Tiangen (TIANGEN Biotech (Beijing) Co., Ltd., Beijing, China). Tryptone and yeast extract were purchased from Thermo (Thermo Fisher Scientific Inc., Waltham, MA, USA). *E. coli* DH5α and *E. coli* BL21 (DE3) were purchased from Tiangen. Z/B/L represent the BL21 (DE3) harboring pACCAR25ΔcrtX, pACCAR16Δcrt, or pACCRT-EIB, respectively. All the chemical reagents utilized in this experiment were of analytical grade.

### 4.2. Plasmid Construction and Culture Conditions

The *Cs*CCD2 (Genbank Accession Number: KJ541749.1) gene with a single-base mutation G1179T from *Crocus sativus* was synthesized by GenScript (GenScript Biotech Corp., Nanjing, China) and subcloned into a pET32a vector using KpnI and EcoRI restriction endonucleases (the G1179T single-base mutation is a synonymous mutation that aims to remove an EcoRI site from the coding sequence). In order to optimize *Cs*CCD2 protein soluble expression, pET41a-GST-F/R, pET28a-SUMO-F/R, and pET28a-MBP-F/R primers were used for the cloning of *Cs*CCD2, which carried corresponding restriction sites (Table 2). Next, the *Cs*CCD2 was subcloned into the pET41a (with GST tag) and the modified pET28a (with SUMO tag or MBP tag) prokaryotic expression vectors using digestion and ligation. To further delve into the reasons behind the non-conversion of DAN1-*Cs*CCD2, a truncated mutation study was carried out. CCD2-T-F and CCD2-T-R were used for the truncation study of CsCCD2 using one-step cloning. All the recombinant plasmids were confirmed using colony polymerase chain reaction and Sanger sequencing. *E. coli* DH5α was used for plasmid propagation and *E. coli* BL21 (DE3) was used for the expression of recombinant protein. Lysogeny broth medium (per liter: 10 g tryptone, 5 g yeast extract, and 10 g NaCl) was used for the propagation of *E. coli*.

### 4.3. The Function Assay of CsCCD2 in Bacteria

The TRX-*Cs*CCD2 vector was co-transformed with pACCAR25ΔcrtX, pACCRT-EIB, or pACCAR16Δcrt into *E. coli* BL21 (DE3). The engineered strains were precultured overnight, and 2 mL cultures were inoculated into 40 mL LB with 50 μg/mL ampicillin and 34 μg/mL chloramphenicol. After being grown at 37 °C and 200 rpm for approximately 3 h (OD_600_ about 0.6), 1.0 M IPTG was added for a final concentration of 0.3 mM, and the culture was induced at 16 °C and 160 rpm for 24 h. The cultures were centrifuged at 4500× *g* for 10 min, and the pellets were ultrasonically extracted with 800 μL acetone to detect the catalyst. The extract was filtered into vials for UPLC and UPLC-MS/MS analysis. Prof. Giovanni Giuliano generously contributed the plasmids pTHIO-DAN1 and DAN1-*Cs*CCD2. Their function assays were repeated as described [8]. Following co-transformation of pTHIO-DAN1 and DAN1-*Cs*CCD2 with pACCAR25ΔcrtX, pACCRT-EIB, or pACCAR16Δcrt into *E. coli* BL21 (DE3), the positive colony was cultured overnight. Two milliliters of cultures were inoculated into 50 mL of LB (containing 50 μg/mL ampicillin and 34 μg/mL chloramphenicol) and grown at 37 °C for 3 h to an OD_600_ of 0.7. The cells were induced using 0.2% (wt/vol) arabinose for 16 h at 20 °C. Moreover, the fusion proteins (GST-*Cs*CCD2, SUMO-*Cs*CCD2, and MBP-*Cs*CCD2) were induced following the protocol of TRX-*Cs*CCD2 induction. Truncated protein (DAN1-*Cs*CCD2-T) was examined based on the methods of DAN1-*Cs*CCD2 induction. 

### 4.4. UPLC and LC-MS/MS Analysis of the Product

The samples were analyzed using a Thermo Ultimate 3000 system (Thermo Fisher Scientific Inc., Waltham, MA, USA) equipped with a Waters Acquity UPLC^®^ BEH C18 column (1.7 μm, 100 × 2.1 mm, Waters Co., Milford, MA, USA) at 35 °C with a wavelength detector set to 440 nm. The mobile phases of acetonitrile comprising 0.1% formic acid (A) and water comprising 0.1% formic acid (B) were used for UPLC. At a flow rate of 0.2 mL/min, the following gradient elution program was used: 0–8 min, linearly increasing from 10% A to 50% A; 8–12 min, linearly increasing from 50% A to 90% A; 12–13 min, linearly increasing from 90% A to 100% A; sustained 30 min.

The qualitative analysis of each product was performed using Agilent Technologies 1290 Infinity II and 6545 Q-TOF, together with Dual Agilent Jet Stream Electrospray Ionization sources (Agilent Technologies, Santa Clara, CA, USA). The drying gas was set at 350 °C and 8 L/min; the sheath gas was set at 350 °C, with a gas flow rate of 11 L/min. The nebulizer was set at 35 PSIG; the VCap was set at 3500 V. The data were analyzed using MassHunter (version B.07.00).

### 4.5. Optimization of CsCCD2 Protein Expression Pattern

The fusion expression strategy was adopted to enhance the soluble expression of *Cs*CCD2. Three expression vectors, including GST-*Cs*CCD2, SUMO-*Cs*CCD2, and MBP-*Cs*CCD2, were constructed as aforementioned. Furthermore, various inducing temperatures (37 °C, 28 °C, or 16 °C) and inducer concentrations (0.1 mM, 0.3 mM, 0.5 mM, 0.8 mM, 1.0 mM, 1.5 mM, and 2.0 mM) were also examined. When the OD_600_ of engineered strains reached 0.6, 0.3 mM IPTG was added to induce protein expression at 160 rpm for 24 h. SDS-PAGE was used to evaluate the protein expression level. Subsequently, the function of fusion *Cs*CCD2 proteins was confirmed using an in vivo assay, and the product was detected as previously stated.

### 4.6. The Bioinformatic Analysis of CsCCD2, TRX-CsCCD2, and DAN1-CsCCD2

The ORFs of *Cs*CCD2, TRX-*Cs*CCD2, and DAN1-*Cs*CCD2 were precisely identified and translated into amino acids using MEGA (Version 6). Their amino acid sequences were further subjected to multiple sequence alignment via DNAMAN (Version 6) to predict differences in sequence. The common characteristics of the ORFs were predicted using SnapGene (Version 4.1.9). The three-dimensional structure of *Cs*CCD2 was predicted using the online Robetta service (https://robetta.bakerlab.org/, accessed on 22 May 2023).

## 5. Conclusions

In summary, this study demonstrated that *Cs*CCD2, a first key enzyme in saffron crocin biosynthesis, can catalyze not only the cleavage of zeaxanthin but also the cleavage of β-carotene and lycopene. The substrate profile of *Cs*CCD2 has been expanded to allow for the use of less-expensive raw materials (especially β-carotene) in crocin synthesis via substrate-feeding strategies. Additionally, the soluble expression of *Cs*CCD2 was determined using *E. coli* as the host organism, paving the path for *Cs*CCD2 crystal structure resolution and facilitating the synthesis of crocetin and crocins using microbial fermentation.

## Figures and Tables

**Figure 1 ijms-24-15090-f001:**
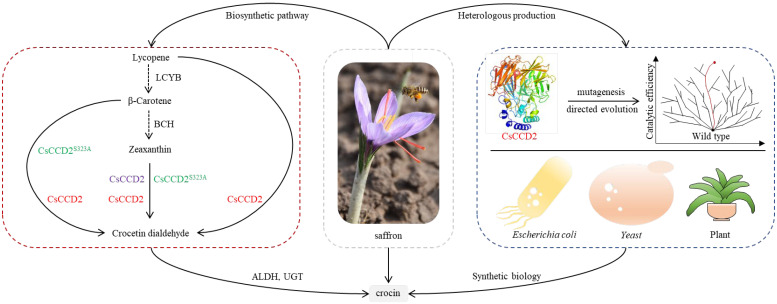
The function identification of *Cs*CCD2 in saffron and its application in synthetic biology. Note: The different color of *Cs*CCD2 represent different results reported in PNAS (purple), J Agric Food Chem (green) and this study (red).

**Figure 2 ijms-24-15090-f002:**
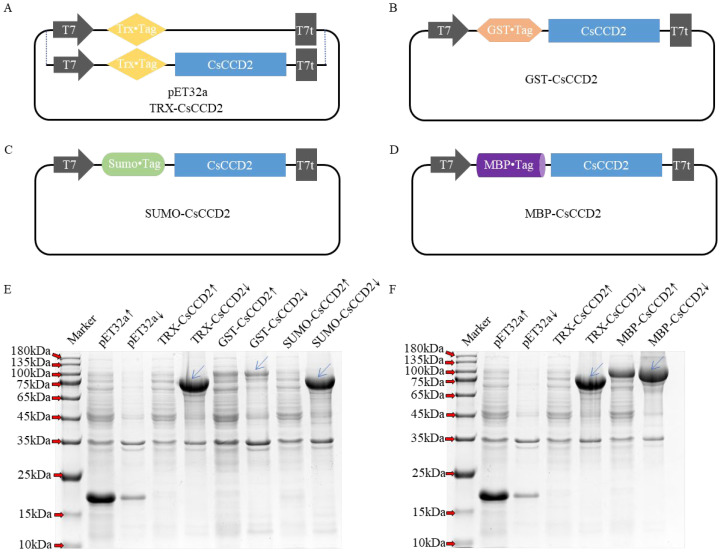
Different expression patterns plasmid maps and its protein expression analysis. (**A**–**D**), five plasmid maps of pET32a, TRX-*Cs*CCD2, GST-*Cs*CCD2, Sumo-*Cs*CCD2 and MBP-*Cs*CCD2. (**E**,**F**), SDS-PAGE analysis of the *Cs*CCD2 with different fusion tags at 16 °C. ↑, indicates the protein supernatant. ↓, indicates the protein precipitate. The band of *Cs*CCD2 fusion protein was labeled with blue arrows in (**E**,**F**).

**Figure 3 ijms-24-15090-f003:**
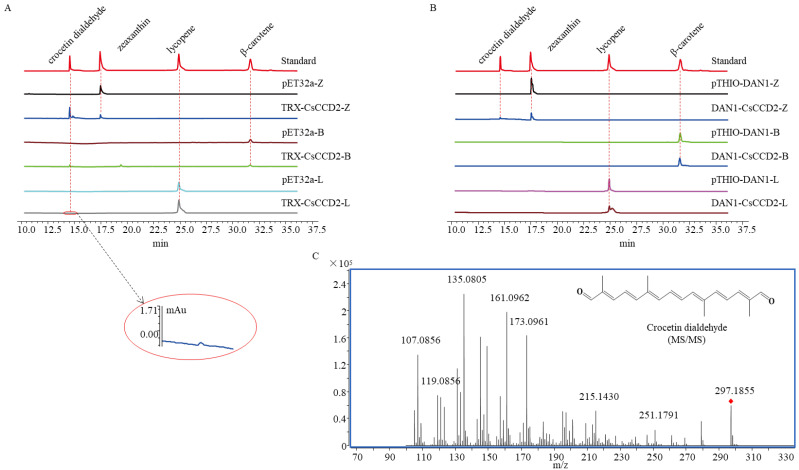
Functional identification of *Cs*CCD2. (**A**), UPLC-DAD chromatograms (abs at 440 nm) of pET32a-Z/B/L and TRX-*Cs*CCD2-Z/B/L extracts. (**B**), UPLC-DAD chromatograms (abs at 440 nm) of pTHIO-DAN1-Z/L/B and DAN1-*Cs*CCD2-Z/L/B extracts. (**C**), the MS/MS fragmentation pattern of crocetin dialdehyde. The red diamond represents the [M+H]^+^ of crocetin dialdehyde.

**Figure 4 ijms-24-15090-f004:**
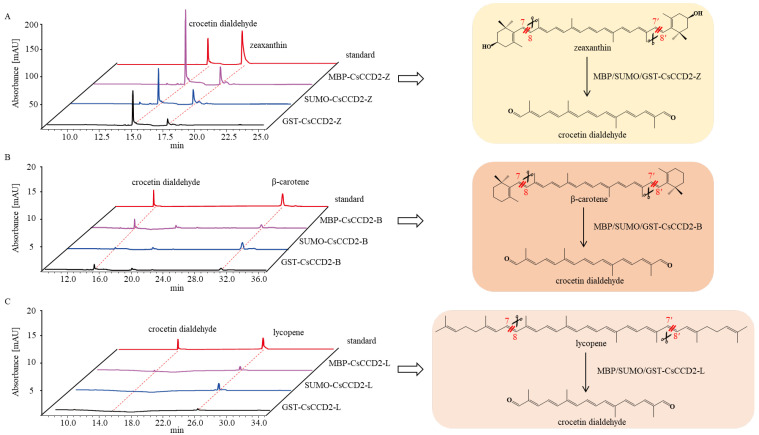
Functional characteristic of *Cs*CCD2-fusion protein. (**A**), UPLC detection of MBP/SUMO/GST-*Cs*CCD2-Z extracts at 440 nm. (**B**), UPLC detection of MBP/SUMO/GST-*Cs*CCD2-B extracts at 440 nm. (**C**), UPLC detection of MBP/SUMO/GST-*Cs*CCD2-L extracts at 440 nm.

**Figure 5 ijms-24-15090-f005:**
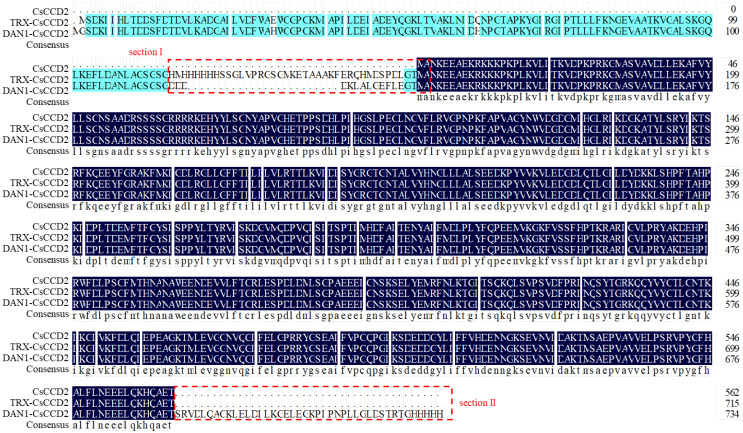
Amino acid sequence comparative analysis of *Cs*CCD2, TRX-*Cs*CCD2 and DAN1-*Cs*CCD2.Dark blue represents the same amino acid sequence among *Cs*CCD2, TRX-*Cs*CCD2 and DAN1-*Cs*CCD2. Wathet blue indicates the same amino acid sequence between TRX-*Cs*CCD2 and DAN1-*Cs*CCD2.

**Figure 6 ijms-24-15090-f006:**
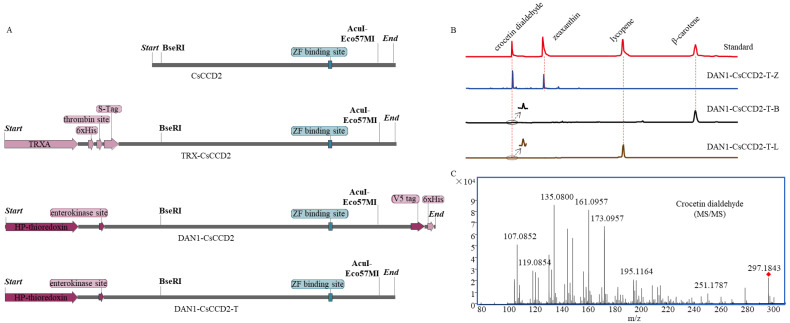
The truncation study of *Cs*CCD2. (**A**), the common features of *Cs*CCD2 ORF. (**B**), UPLC-DAD chromatograms (abs at 440 nm) of DAN1-*Cs*CCD2-T-Z/B/L extracts. (**C**), the MS/MS fragmentation pattern of crocetin dialdehyde. The red diamond represents the [M+H]^+^ of crocetin dialdehyde.

**Table 1 ijms-24-15090-t001:** Plasmids used in this study.

Plasmids	Relevant Properties or Genetic Marker ^a^	Source or Reference
pACCAR25ΔcrtX	pACYC184 plus crtE, crtI, crtB, crtY, and crtZ from *E. uredovora*, Cm^R^	[26]
pACCAR16Δcrt	pACYC184 plus crtE, crtI, crtB, and crtY from *E. uredovora*, Cm^R^	[26]
pACCRT-EIB	pACYC184 plus crtE, crtI, and crtB from *E. uredovora*, Cm^R^	[26]
pTHIO-DAN1	*pBR322* ori and *pUC* ori, Amp^R^	[27]
DAN1-*Cs*CCD2	pTHIO-DAN1 plus *Cs*CCD2 from saffron	[8]
DAN1-*Cs*CCD2-T	pTHIO-DAN1 plus truncated *Cs*CCD2	This study
pET32a	*pBR322* ori and *f1* ori, Amp^R^	Novagen
TRX-*Cs*CCD2	pET32a plus *Cs*CCD2 from saffron	This study
pET41a	*pBR322* ori and *f1* ori, Kan^R^	Novagen
GST-*Cs*CCD2	pET41a plus *Cs*CCD2 from saffron	This study
pET28a	*pBR322* ori and *f1* ori, Kan^R^	Novagen
pET28a-SUMO	pET28a plus SUMO•Tag	This study
pET28a-MBP	pET28a plus MBP•Tag	This study
SUMO-*Cs*CCD2	pET28a-SUMO•Tag plus *Cs*CCD2 from saffron	This study
MBP-*Cs*CCD2	pET28a-MBP•Tag plus *Cs*CCD2 from saffron	This study

^a^ Amp^R^, Kan^R^, and Cm^R^ represent ampicillin, kanamycin, and chloramphenicol, respectively.

**Table 2 ijms-24-15090-t002:** Primers used in this study.

Primers	Sequence (5′–3′)	Restriction Endonuclease (Underlined)
pET41a-GST-F	CATGCCATGGGCGAAAACCTGTACTTTCAAGGCATGGCAAATAAGGAGGAG	NcoI
pET41a-GST-R	CCCAAGCTTTCATGTCTCTGCTTGGTGCTTCTG	HindIII
pET28a-SUMO-F	CCCAAGCTTCCGAAAACCTGTACTTTCAAGGCATGGCAAATAAGGAGGAG	HindIII
pET28a-SUMO-R	AAGGAAAAAAGCGGCCGCTCATGTCTCTGCTTGGTGCTTCTGAAGTTC	NotI
pET28a-MBP-F	CTAGCTAGCGAAAACCTGTACTTTCAAGGCCATATGATGGCAAATAAGGAGGAGGCAG	NheI
pET28a-MBP-R	CCCAAGCTTTCATGTCTCTGCTTGGTGCTTCTG	HindIII
CCD2-T-F	CCAAGCAGAGACATGAGTTTAAACGGTCTCCAGCTT	-
CCD2-T-R	ACTCATGTCTCTGCTTGGTGCTTCTGAAGTTCT	-

## Data Availability

Not applicable.

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
