# Peer review of "The Functional Characteristics and Soluble Expression of Saffron *Cs*CCD2"

_ijms, 2023, doi:10.3390/ijms242015090_

Round 1

Reviewer 1 Report

The paper from Wang et al. is focused on the production of saffron Carotenoid cleavage dioxygenase 2 (CCD2) in E. coli and its activity on three different substrates. Although this has been done before (ref 8 in the paper), the efficiency was low, due to the insoluble form of the protein. To overcome this limitation, the authors have tried various fusion tags together with various induction conditions. The use of the MBP tag and a low temperature of induction resulted in the production of more soluble protein, translating into higher enzyme activity in bacterio. This higher enzyme activity might explain the detection of a reaction product from all three substrates, instead of only zeaxanthine as found before. The authors demonstrated here that this restriction was possibly due to the presence of an extension at the C-terminal end of the enzyme produced before. The manuscript is well presented and globally well written but there are a number of issues that could be addressed to enhance the accessibility of the results.

- Details of the protocol (lines 23-25) should be avoided in the abstract and replaced by a more general statement. For example: "the use of a maltose-binding protein (MBP) tag and the op5mization of the induction conditions resulted in the production of more soluble protein".

The use of case numbers to describe the experiments using the various tags makes it difficult to visualize the results in the figures. It would be better to replace "case" by the names of the tags: TRX, GST, SUMO, MBP. This should be done in figures 2E, 2F and 4.

On images 2E and 2F, the sizes of all the marker bands should be indicated.

Table 1 is hard to read because of the line shifts between the text in column 1 and 2. There is also too much repetition in the second part of this table. For clarity it would be better to add in the list of plasmids, the pACC... encoding Z, L and B. The use of "strains" is improper here because there is only one strain: BL21(DE3) with various plasmids. This is why the second sub-sections of this table could be removed. The BL21(DE3) genotype could be given in the material section.

Line 45, "domestic and foreign counterparts" is irrelevant in an international journal.

Line 122-123: replace sentence by: "Additionally, different induction temperatures and IPTG concentrations were tested."

Line 149: replace "has" by "can have".

Line 183: ... splibng activity on...

The sentence lines 198-201 could be removed.

In figure 6A, it would be nice to have the plasmid name indicated under each map.

The method section falls short on some aspects. The details of the cloning procedures should be given. For example, line 226, how was the cloning into pET41a and pET28a done? How were the oligos of Table 2 used for the cloning?

Was the single base mutation in the CsCCD2 sequence (line 224) introduced to create the S323A amino acid change mentioned line 70? If so, it has to be explained.

Line 246, what is meant by: "the function characteristics of them were repeated"?

It has to be stated in the introduction that the plasmids used to make E. coli produce lycopene, zeaxanthin or ß-carotene contains genes from Erwinia uredovora encoding several enzymes from the carotenoid biosynthesis pathway.

Moderate editing of English language required.

Reviewer 2 Report

The manuscript details the optimization of heterologous expression conditions for CCD2 from saffron and the confirmation of CsCCD2's catalytic functionality.

Results indicate that CsCCD2 was effectively expressed in its soluble form using E. coli as the host organism. Intriguingly, beta-carotene and lycopene, previously believed to be non-recognizable as substrates, were indeed accepted as substrates. The authors further delved into the reasons behind the non-conversion of these compounds in the past study by conducting mutation analysis. This research significantly advances the efficient production of industrially valuable crocins.

While the findings are presented in a lucid manner, and the methodology and results are succinct and articulate, there are concerns regarding some of the figures. A few are crowded and minuscule, making it challenging for readers to discern. It is recommended that the authors enhance the clarity of these figures, either by providing magnified versions in the supplementary materials or by amplifying the size of the chromatograms and reducing the size of other ancillary details.

Moreover, the use of the term "neofunctionalization" in the title seems misplaced. Typically, "neofunctionalization" refers to the evolutionary acquisition of a novel function. In this study, however, a previously misunderstood original function was confirmed, rendering the term inappropriate.

With these modifications, the manuscript appears suitable for acceptance.

Round 2

Reviewer 1 Report

The issues of the first draft have been addressed. One correction has to be made:

Lines 269-270: the G1179T point mutation was not "to prevent the degradation of EcoRI restriction endonuclease" as stated but "to remove an EcoRI site from the coding sequence". 
